# Update: Immunotherapeutic Strategies in HPV-Associated Head and Neck Squamous Cell Carcinoma

**DOI:** 10.3390/v17050712

**Published:** 2025-05-16

**Authors:** Fangdi Sun, A. Dimitrios Colevas

**Affiliations:** Division of Oncology, Department of Medicine, Stanford University, Stanford, CA 94305, USA; sunf@stanford.edu

**Keywords:** human papillomavirus (HPV), head and neck cancer, oropharyngeal cancer, squamous cell carcinoma, immunotherapy, immune checkpoint inhibitor, adoptive T-cell therapy

## Abstract

The incidence of human papillomavirus (HPV)-associated oropharyngeal squamous cell carcinoma (OPSCC) has increased substantially over the past three decades, and since 2017, it has been recognized in the AJCC staging system as distinct from its HPV-negative counterpart. The underlying mechanisms of HPV-associated carcinogenesis, tumor microenvironment, and host immune response represent opportunities for therapeutic development. While anti-PD-1 immunotherapy is now part of standard treatment for recurrent or metastatic head and neck squamous cell carcinoma (HNSCC) in general, there are no established immunotherapeutic strategies specifically for HPV-related HNSCC. In this context, multiple emerging approaches are being actively studied—among these are therapeutic vaccines with or without anti-PD-(L)1 adjuvants, peptide–HLA-based immunotherapeutic platforms, and adoptive cell therapies including tumor-infiltrating lymphocytes (TILs), T-cell receptor (TCR) therapy, and chimeric antigen receptor (CAR) T-cell therapy. Beyond further maturation of these novel immunotherapeutic strategies, additional work is needed to delineate the optimal disease state of application (localized versus recurrent/metastatic), as well as in the development of small molecule inhibitors targeting HPV-specific mechanisms of viral oncogenesis.

## 1. Introduction

In 2022, cancers of the head and neck accounted for 950,000 new cases and 480,000 cancer deaths globally, making up approximately 5% of both new cases and deaths worldwide [1]. Malignancies of epithelial origin, specifically squamous cell carcinoma (SCC), comprise the majority of cancers identified at the major anatomical sites: oral cavity, pharynx (including the oropharynx, hypopharynx, and nasopharynx), larynx, and nasal cavity and paranasal sinuses (also called the sinonasal tract) [2,3,4]. For the purposes of clarity, most cancers of the nasopharynx are Ebstein–Barr virus (EBV)-associated, which is considered a distinct entity from most head and neck SCC (HNSCC) [5]. In the United States, nearly two-thirds of HNSCC cases are locally advanced or metastatic at diagnosis [6].

Traditionally, HNSCC has been considered a disease of older men, associated with major risk factors of heavy tobacco and alcohol use. The incidence of tobacco-associated cancers, including human papillomavirus (HPV)-negative HNSCC, has been decreasing in the United States, Europe, and other developed countries, whereas it continues to rise in developing countries, including much of Asia [4,7]. By contrast, particularly in developed countries, rates of human papillomavirus (HPV)-associated HNSCC have increased substantially over the last three decades. While HPV has been recognized as a potential etiologic factor in HNSCC since the 1980s, increased provider recognition, directed testing, and potential changes in oral sexual practices have led to a substantial increase in the proportion of head and neck cancers that are HPV-associated over this period [8,9,10]. Though a vaccine-preventable disease, public awareness of HPV as a risk factor for oropharyngeal cancer remains poor [11].

In this review, we will discuss HPV-associated HNSCC, focusing on the underlying mechanisms of HPV-associated carcinogenesis, the tumor microenvironment, and the host immune response as potential susceptibilities for therapeutic development. We will review the current standard of checkpoint inhibitor immunotherapy in the recurrent or metastatic setting, focusing on outcomes in HPV-associated disease. While there are presently no established immunotherapeutic strategies specifically for HPV-related SCC, we will discuss several emerging immunotherapeutic approaches (including vaccines and adoptive T-cell therapies) in clinical testing and end with commentary on future directions.

## 2. HPV-Associated Oropharyngeal Squamous Cell Carcinoma (OPSCC)

Infection with HPV accounted for approximately 30% of new oropharyngeal carcinoma cases worldwide in 2018 [12]. In recent decades, this HPV-attributable fraction has been consistently higher among high-income countries, including much of Europe and North America, compared to lower-income regions [12,13]. HPV detection in oropharyngeal cancers increased from 16 to 72% between the 1980s and early 2000s in the United States, and was reported at 52% in the United Kingdom between 2002 and 2011 [10,14]. While HNSCC of non-oropharyngeal sites can be associated with HPV infection at reported rates ranging from 2 to 24%, co-testing for p16 expression by immunohistochemistry (IHC) is less commonly performed, making causality and thus functional significance harder to assess [12,15,16]. Though there are over 230 fully characterized types of HPV, 14 are considered the highest risk for HPV-associated carcinogenesis [17,18,19]. HPV-16 is the highest risk mucosal subtype, identified in 85 to 90% of HPV-associated oropharyngeal squamous cell carcinomas (OPSCCs). Other high-risk HPV subtypes, including HPV-18 and HPV-33, are identified much less commonly, with typical frequencies of less than 1 or 2% [20]. Notably, this HPV distribution differs from cervical cancer, where approximately half of cases are attributed to HPV-16 [21].

The recognition of HPV as a key etiologic factor in OPSCC led to the observation of substantially better clinical outcomes among patients with HPV-associated disease compared to those whose cancers were HPV-negative [22,23]. Therefore, in 2017, the American Joint Committee on Cancer (AJCC) updated its eighth edition staging manual to include a separate staging system for HPV-associated OPSCC. This resulted in an overall downstaging of HPV-associated disease according to the TNM criteria relative to the previous seventh edition and relative to HPV-negative OPSCC [24,25]. HPV status does not affect staging or treatment decisions at non-oropharyngeal sites and may not be routinely assessed. Most patients with clinically localized HPV-associated OPSCC are treated with definitive concurrent chemoradiotherapy, though surgery or radiation alone are often used for small (cT1-2) tumors, particularly those that are node-negative (cN0) [2]. In aggregates, approximately two-thirds of patients with localized HNSCC will develop disease recurrence, typically within two years [3]. While outcomes are significantly better for HPV-associated disease, around 25 to 30% of patients will still recur [26,27]. For these cases of metastatic disease or recurrence which cannot be managed with salvage local therapies, the preferred first-line regimen is either pembrolizumab monotherapy (for PD-L1 CPS ≥ 1) or pembrolizumab with platinum-doublet chemotherapy (any PD-L1) based on KEYNOTE-048 [28]. While a small minority of patients experience prolonged disease control with checkpoint inhibitor immunotherapy, median PFS remains approximately one year, and options for subsequent systemic treatment remain an area of unmet need [29].

## 3. Molecular Biology of HPV-Associated Carcinogenesis

HPVs are non-enveloped, double-stranded circular deoxyribonucleic acid (DNA) viruses that infect the human squamous epithelium. The viral genome consists of approximately 8000 base pairs within which are three functional regions—a cluster of “early” viral genes (E1–E7) required for replication, a set of “late” genes (L1, L2) required for viral capsid formation, and a large non-coding long control region (LCR) [30,31]. Within the infected host cell, HPV E6 and E7 viral proteins are primarily implicated in carcinogenesis. Via ubiquitin-mediated processes, E6 and E7 function chiefly to induce degradation of the tumor suppressor proteins p53 and retinoblastoma (pRb), respectively [32]. Downstream of pRb degradation, E2F promotes the transcription of *CDKN2A* and consequently increases p16^INK4a^ expression [33]. Overall, this leads to triggering cell cycle synthesis (S) phase re-entry on top of loss of p53-dependent cell cycle arrest, thereby promoting further viral DNA replication [34]. This simplified version of HPV pathogenesis illustrates the diagnostic tests utilized in clinical practice. IHC for p16 expression is most common due to its low cost and rapid turnaround, and the eighth edition of the AJCC staging system defines HPV association based on p16 status [24]. While highly sensitive for HPV-associated OPSCC, p16 IHC has only moderate specificity, leading to the recommendation that p16 IHC be combined with HPV DNA PCR to improve this [35]. While detection of E6/E7 mRNA is considered the gold standard diagnostic assay, it is infrequently used due to limited availability and cost [36].

In addition to the described mechanism via p53 and pRb, the E6 and E7 oncoproteins have other cellular protein targets which promote carcinogenesis. For example, E6 can lead to the degradation of PDZ domain-containing proteins, many of which function as tumor suppressors and have critical roles in signal transduction [37]. E6 has also been shown to promote NFX1-91 degradation, removing it from the hTERT promoter and leading to increased telomerase expression [38,39]. Class I histone deacetylases (HDACs) may be downstream effectors of E7 via E2F-induced transcription, leading to changes in chromatin remodeling and gene expression [37,40,41].

The E1 and E2 viral proteins are involved in viral replication and gene expression. E1 is the only HPV viral protein product with direct enzymatic function and acts as a DNA helicase [42]. E2 is critical for the regulation of viral gene transcription, as well as the segregation of viral plasmids during cell division. In HPV-associated malignancies, the viral genome may be episomal and/or integrated into the host cell genome. Integration often occurs in the E1 or E2 region and can lead to loss of negative feedback from the E2 protein [43]. HPV integration may be associated with inferior prognosis and more aggressive clinical phenotypes in HNSCC; however, conflicting studies have also been reported, perhaps a consequence of variable detection methods to identify integration events [43,44,45,46].

The overall genomic landscape of HPV-associated HNSCC has been compared to that of its HPV-negative counterpart. HPV-associated tumors are consistently associated with mutations in *PIK3CA*, with mutations in *DDX3X*, *FGFR2/3*, and *KRAS* variably reported. In contrast, HPV-negative tumors are enriched for mutations in *TP53* and *CDKN2A* and more frequently exhibit amplifications in *EGFR*, *CCND1*, and *FGFR1* [47,48,49,50]. Overall tumor mutational burden (TMB) is comparable between HPV-associated and HPV-negative HNSCC, though TMB is higher among HPV-associated tumors in patients with significant smoking history (≥10 pack-years) [48,50].

## 4. Immune System Evasion and Tumor Microenvironment in HPV-Associated HNSCC

HPVs have developed a number of mechanisms mediating immune evasion by HPV-associated malignancies. T-cell activation relies on intact antigen presentation by antigen-presenting cells (APCs) complexed with major histocompatibility complex (MHC) class I and II molecules, the human version of which is the human leukocyte antigen (HLA) system [51]. A large-scale genome-wide association study (GWAS) has identified multiple HLA loci associated with oropharyngeal cancer risk, with statistical significance among only the HPV-associated cancers [52]. In vitro, both the HPV-16 and HPV-18 E7 proteins lead to repression of the MHC class I heavy chain promoter [53]. The HPV E5 protein has been shown to downregulate peptide-bound MHC molecules via the alkalinization of late endosomes, thereby limiting the activation of antitumor T-cells [54,55]. In particular, HPV selectively downregulates HLA-A, HLB-B, and HLA-C molecules but spares the nonclassical HLA-E molecule, functioning as an inhibitory signal for cytotoxic natural killer (NK) cell activity [54].

The cGAS-STING pathway is a conserved component of the innate immune system that senses cytosolic DNA and serves as a critical defense mechanism against DNA viruses [56]. The activation of this pathway by viral antigens leads to a cytokine milieu and tumor microenvironment (TME) favoring a cytotoxic T-cell response and viral clearance [57]. Though much of the data remain preclinical, HPV E7 has been shown to downregulate the cGAS-STING pathway though various mechanisms [58,59].

Expression of the PD-L1 immune checkpoint protein has been reported at higher frequencies in patients with HPV-associated compared to HPV-negative tumors [60,61]. However, large cohort analyses have demonstrated that HPV association predicts better response to checkpoint inhibitor immunotherapy independent of PD-L1 status [62]. These data, combined with other studies looking at concomitant tumor-infiltrating lymphocyte (TIL) infiltrate and gene expression profiles, suggests that PD-L1 expression may be a proxy for a generally inflamed TME rather than a direct effect of HPV [57,60].

In spite of these mechanisms of immune evasion, there are a number of HPV-specific immune responses, many of which are associated with prognostic significance. HPV-16 E6- and E7-specific T-cells have been identified in both tumor and blood samples of patients with HPV-16-associated HNSCC [63,64]. These TILs are skewed towards CD4^+^ T-cells with an antitumor T helper (T_H_)1/T_H_17 profile [63]. More generally, higher levels of TILs have been demonstrated in HPV-associated disease compared with HPV-negative HNSCC and are correlated with better clinical response [65,66,67]. Furthermore, high numbers of CD4^+^ FOXP3^+^ regulatory T-cells have been identified in HNSCC and are associated with inferior outcomes, though the relative prevalence in HPV-associated versus HPV-negative disease is presently unclear [64]. Factors involving other immune cell types, including lower CD20^+^ B-cells, lower dendritic cell (DC)-like macrophages, higher tumor associated macrophages (TAMs), and higher myeloid-derived suppressor cells (MDSCs), have also been associated with inferior clinical outcomes but are not necessarily HPV-specific [63,68,69,70,71,72,73].

## 5. Checkpoint Inhibitor Immunotherapy

Given this background, there is biologic rationale for checkpoint inhibitor immunotherapy in HPV-associated HNSCC. In reality, PD-(L)1 inhibitors have largely been studied in HPV-agnostic cohorts, with only secondary analyses of efficacy by HPV status. KEYNOTE-048 was the practice changing, randomized phase III trial investigating pembrolizumab alone or with chemotherapy versus the established EXTREME regimen of cetuximab plus chemotherapy in the front-line, recurrent, or metastatic settings. Overall survival (OS), but not PFS or objective response rate (ORR), was improved with pembrolizumab alone in the PD-L1 CPS ≥ 1 population (12.3 vs. 10.3 months, *p* = 0.009) and for pembrolizumab plus chemotherapy in all comers (13.0 vs. 10.7 months, *p* = 0.003), both in comparison to cetuximab plus chemotherapy. Of the 447 patients with oropharyngeal primary tumors, 56% were p16-positive by IHC [28]. In the four-year follow-up, the magnitude of OS benefit appeared similar between the p16-positive and p16-negative subgroups [29].

A number of studies investigating PD-(L)1 inhibition in recurrent or metastatic HPV-associated OPSCC after progression on platinum-based regimens report numerically improved response rates or overall survival for HPV-associated cancers compared to HPV-negative cancers [74,75,76,77,78]. Other studies have not reported such differences [79,80]. However, these trends should be interpreted cautiously, as many of these analyses report point estimates from small subgroups, leading to wide confidence intervals and an inability to demonstrate statistical significance. Study design varied across trials, with specific inclusion/exclusion criteria and variation in PD-L1 expression status potentially leading to differential treatment response. Furthermore, a pooled analysis of nine clinical trials of PD-(L)1 inhibitor monotherapy in advanced HNSCC demonstrated numerically higher but not statistically significant ORR (18.8 vs. 12.2%) and DCR (42.8 vs. 34.4%) for HPV-associated disease compared to HPV-negative disease [81]. Ultimately, PD-(L)1 inhibition has demonstrated benefit in both entities, which are biologically and molecularly distinct, as previously emphasized.

Antibodies targeting immune checkpoints other than PD-(L)1 have been studied in basket trials of advanced solid tumors and dedicated cohorts of HNSCC. The PD-L1 inhibitor durvalumab in combination with the CTLA-4 inhibitor tremelimumab has been studied in both the first-(CONDOR, EAGLE) and subsequent-line settings (KESTREL) for advanced HNSCC, with no differences in efficacy compared to standard therapy and no evident difference by HPV status [82,83,84]. Other immune checkpoints including TIM-3, TIGIT, LAG-3, and VISTA are areas of active investigation, recently reviewed by Struckmeier et al. [85]. Data from these studies remain preliminary, oftentimes with monoclonal antibodies given in combination with PD-(L)1 inhibitor immunotherapy or as bispecific antibodies that target both PD-(L)1 and the other immune checkpoint [85]. Whether these novel checkpoint inhibition strategies have differential efficacy by HPV status has yet to be determined.

EGFR inhibitors (namely cetuximab) have also been studied in combination with the PD-1 inhibitors pembrolizumab or nivolumab. A recent meta-analysis included 118 patients with advanced HNSCC receiving combination therapy and 684 patients receiving PD-1 inhibitor monotherapy. Pooled ORR and OS were significantly improved for combination cetuximab and PD-1 inhibition compared to PD-1 inhibitor monotherapy in the HPV-negative subgroup (ORR 46 vs. 15%, *p* < 0.001; 1-year OS rate 59 vs. 36%, *p* < 0.001), but not in those with HPV-associated disease (ORR 18 vs. 17%, *p* = 0.686; 1-year OS rate 55 vs. 40%, *p* = 0.252) [86]. While these results require further validation in prospective studies, the aforementioned enrichment of EGFR pathway activation in HPV-negative HNSCC lends biological credence to this observation [48,87].

## 6. Therapeutic HPV Vaccines

While prophylactic vaccines derived from viral L1 proteins are widely recommended for the prevention of HPV-associated malignancy, different targets are needed for therapeutic vaccination [88,89]. In general, the goal of therapeutic vaccination is to stimulate an adaptive immune response capable of inducing cytotoxic activity with sufficient sensitivity and specificity to allow for tumor control while minimizing adverse effects. HPV oncoproteins E6 and E7 have long been considered ideal target “non-self” antigens because they are exclusively produced by cancer cells, constitutively expressed, and persistently necessary for oncogenesis [90,91,92]. Several cancer vaccine platforms have emerged in the HPV-associated HNSCC space, broadly divided into molecular vaccines (DNA-, RNA-, or peptide-based) and viral or bacterial vector-based vaccines [92]. All of these strategies are designed to induce a host immune response against the target antigen and thus the cancer cell, but do not directly interfere with E6 or E7 function.

DNA vaccines on their own are poorly immunogenic in humans; therefore, many constructs incorporate cytokine coding genes or are co-administered with anti-PD-(L)1 immunotherapy [93]. MEDI0457 is a DNA vaccine encoding the E6 and E7 antigens from HPV-16 and 18, administered together with a plasmid encoding an IL-12 adjuvant. As a monotherapy, MEDI0457 was studied in locally advanced, p16-positive HNSCC. This study demonstrated antigen-specific T-cell activity and cellular responses exceeding one year but was not designed to evaluate clinical efficacy [94]. MEDI0457 was subsequently studied in combination with the PD-L1 inhibitor durvalumab in a Phase Ib/IIa trial, including 29 evaluable patients with previously treated HPV-associated HNSCC who were immunotherapy-naïve. ORR was 27.6% and median PFS 3.5 months; however, the study did not reach its primary efficacy endpoint of ORR (lower bound of 95% confidence interval excludes ORR H_0_ ≤ 15%), likely affected by poor enrollment [95]. AMV002 is a DNA vaccine composed of two plasmids encoding two variants of a fusion protein of E6 and E7, demonstrating enhanced E6- or E7-specific cell-mediated responses in the phase I dose escalation study [96]. In the subsequent phase Ib study in recurrent/metastatic HPV-associated OPSCC, AMV002 with durvalumab was well-tolerated, with no grade 3 or higher vaccine-related adverse events (AEs). There was a single complete response (ORR 8%); however, most patients (83%) in this study were immunotherapy-naïve and the prior treatments and PD-L1 status of the single responding patient were not reported [97]. At present, there are no ongoing studies of either MEDI0457 or AMV002 in HPV-associated HNSCC.

Technologic advancements improving mRNA delivery and stability have made therapeutic RNA vaccines an area of developing interest. BNT113 is a liposome-encapsulated mRNA vaccine encoding for HPV-16 E6 and E7. In the phase I/II study of BNT113 monotherapy across HPV-16-positive cancers of multiple subtypes (including three HNSCCs), disease control was observed in 5/7 and cellular immune responses against E6 or E7 in 3/7 evaluable patients with advanced disease. No objective responses were reported [98]. Preliminary results from the safety run-in phase of the phase II BNT111-01 trial, studying BNT113 plus pembrolizumab as a first-line treatment for recurrent/metastatic HPV-16-positive, PD-L1-positive (CPS ≥ 1) HNSCC have been reported. Among 15 patients, ORR was 40% and median PFS 3.9 months, with treatment-related AEs mainly grade 1–3 flu-like symptoms. The randomized portion of this study (AHEAD-MERIT) is ongoing, comparing the combination to pembrolizumab monotherapy (NCT04534205) [99].

Peptide-based vaccines consist of amino acid sequences containing immunogenic epitopes of the target of interest, typically delivered with a concomitant adjuvant. PDS0101 is a peptide vaccine containing six epitopes of HPV-16 E6 and E7, co-administered with the immune-activating cationic lipid R-DOTAP [100]. In a single-arm study of PDS0101 and pembrolizumab in the first-line setting for PD-L1-positive, recurrent/metastatic HPV-16-positive HNSCC (*n* = 53), ORR was 34% and median PFS 6.3 months [101]. In a separate cohort of checkpoint inhibitor-refractory patients (*n* = 21), no responses were observed [102]. The phase III, randomized VERSATILE-003 study is ongoing, comparing PDS0101 plus pembrolizumab to pembrolizumab monotherapy in the first-line, recurrent/metastatic, PD-L1-positive population (NCT06790966). PDS0101 is also being studied in combination with M9241 (an immunocytokine composed of IL-12 heterodimers) and bintrafusp alfa (a TGF-β “trap”/anti-PD-L1 fusion protein) in advanced HPV-16-positive malignancies (including 13 OPSCC). Interim results show promise, with ORR 88% (7/8) in checkpoint inhibitor-naïve disease and ORR 27% (6/22) in checkpoint-refractory disease [103,104]. ISA101 is composed of overlapping peptide sequences covering the complete HPV-16 E6 and E7 proteins [105]. Building upon single-agent data in cervical cancer, the phase II study of ISA101 with nivolumab enrolled 24 patients with advanced HPV-16-positive malignancy, 22 of whom had OPSCC. ORR was 36% and median PFS 2.5 months, noting that all patients were immunotherapy-naïve [106]. ISA101b is the subsequent iteration of this vaccine, which was studied in the phase II OpcemISA study, randomizing patients with recurrent/metastatic HPV-16- and PD-L1-positive OPSCC (*n* = 198) to either ISA101b plus cemiplimab or cemiplimab monotherapy. The primary outcome of ORR was not different between the arms (25% for combination vs. 23% for control), though an exploratory per protocol analysis among patients with PD-L1 CPS ≥ 20 suggested potential benefit among this subgroup [107].

While live vector-based vaccines are highly immunogenic, the potential safety risks in immunocompromised patients and the induction of neutralizing vector-specific antibodies are common areas of concern [108]. ADSX11-001 is a live *Listeria monocytogenes* vector vaccine engineered to secrete the HPV-16 E7 protein fused to an attenuated *Listeria* virulence factor [109]. A phase I trial of ADSX11-001 in patients with locally advanced HPV-16-positive OPSCC (NCT01598792) was complicated by a case of systemic listeriosis in the second patient enrolled, occurring within 24 h of the first vaccine administration [110]. The trial was paused and ultimately terminated following this severe AE. A separate study in OPSCC prior to robotic surgery reported a few HPV-specific T-cell responses, but results were limited by poor accrual (NCT02002182). HB-200 is a viral vector vaccine composed of two replication-attenuated arenavirus vectors derived from the lymphocytic choriomeningitis (HB-201) and Pichinde viruses (HB-202). It expresses an HPV-16 E7E6 fusion protein and as monotherapy has been shown to induce HPV-16 tumor-specific T-cell responses in heavily pretreated patients [111]. HB-200 has been studied in combination with pembrolizumab as a first-line treatment for HPV-16- and PD-L1-positive recurrent/metastatic HNSCC (*n* = 42), most cases of which were checkpoint inhibitor-naïve (95%). ORR was 43% overall and 59% among PD-L1 CPS ≥ 20 [112]. This study is ongoing and includes a separate cohort for subsequent-line treatment (NCT04180215). PRGN-2009 is a gorilla adenovirus vector vaccine containing multiple epitopes of HPV-16 and 18 E6 and E7, as well as additional T-cell enhancer agonist epitopes [113]. In advanced HPV-16 and 18 cancers, the combination of PRGN-2009 and bintrafusp alfa led to objective response in 3 of 10 evaluable patients (ORR 30%), two of whom were previously refractory to checkpoint inhibitors [114]. PRGN-2009 is also being studied in the neoadjuvant/induction setting, either in combination with chemotherapy or with checkpoint inhibitor immunotherapy (NCT06223568, NCT05996523).

Despite different vaccine platforms, all the aforementioned strategies target HPV E6 and/or E7. Multiple independent groups have demonstrated HPV-specific T-cell responses against epitopes derived from HPV E2 and E5 proteins (in addition to E6 and E7), providing other candidate targets for therapeutic HPV vaccination [115,116,117]. Furthermore, whether personalized vaccines directed against a patient’s tumor-specific antigens could be more effective is unknown, though the appeal of such a strategy may be higher in HPV-negative HNSCC where conserved viral targets do not exist [118]. Table 1 summarizes ongoing studies of therapeutic vaccines in HPV-associated HNSCC.

## 7. Adoptive Cell Therapies

The principle of adoptive cell therapy (ACT) has existed since the 1980s, when the first trials of TILs were first conducted in melanoma [119]. Broadly, the principle of these therapies involves the extraction and expansion of immune cells ex vivo, preparative lymphodepleting chemotherapy to facilitate engraftment, and finally adoptive cell transfer into the patient, with the intent of subsequent in vivo persistence [120,121]. Thus far, most of these therapies involve the manipulation of autologous T lymphocytes, consisting of genetically modified T-cell receptors (TCRs), chimeric antigen receptor (CAR) T-cell therapy, and TILs.

Engineered TCRs involve the modification of a patient’s T-cells to express a TCR recognizing a specific target antigen on cancer cells. As the TCR is engineered to recognize tumor antigens presented by specific MHC molecules, these therapies are HLA-specific, with development thus far largely restricted to HLA-A*02:01. Genetically modified TCRs can be used for both intracellular and cell surface targets, whereas CAR T-cell therapy is restricted to surface antigens because of direct antigen recognition and binding [122]. As with previously discussed vaccination strategies, the intracellular HPV E6 and E7 proteins have been the most common candidate targets, specifically for engineered TCRs. In contrast, the identification of specific and recurrent cell surface antigens for CAR T-cell targeting has thus far limited the development of these therapies specifically for HPV-associated malignancies [57]. Lastly, autologous TIL therapy involves the ex vivo expansion of a heterogenous lymphocyte population, typically derived from a surgical resection specimen. While additional ex vivo selection and genetic modification can be performed, TILs are inherently multi-targeted and therefore less suitable for ubiquitously expressed viral antigens [123].

Only a few clinical studies of ACT in HPV-associated HNSCC have thus far been reported. A phase I/II trial enrolled 12 patients with metastatic HPV-16-positive solid tumors, prior platinum-based chemotherapy, and HLA-A*02:01, including one patient with OPSCC. Patients received lymphodepleting chemotherapy followed by autologous engineered TCRs against HPV-16 E6 with adjuvant high-dose IL-2. Two objective responses were observed at the highest dose level, and no dose-limiting toxicities (DLTs) were reported in the phase I portion. The loss of HLA-A*02:01 and frameshift deletion of the interferon gamma receptor 1 were demonstrated as two potential resistance mechanisms in patients who did not respond to this therapy [124]. Potentially more promising was a phase I study of E7 TCR T-cells in a similar patient population, conducted by the same group. This product resulted in objective responses in 6 of 12 patients, including four of eight with anti-PD-1 refractory disease and two of four with primary head and neck tumors [125]. The phase II component of this study is ongoing (NCT02858310). In general, most ongoing TCR studies focus on E7 TCR T-cells, which in preclinical models have demonstrated higher functional avidity and improved effector function compared to E6 TCR T-cells [126]. Multiple studies of CAR T-cell therapy (and at least one study of CAR NK-cell therapy) are ongoing in HNSCC, though they are not HPV-specific. A phase II trial of the TIL therapy lifileucel (LN-145) has been completed though not published, with an ORR of only 11% among 53 evaluable patients with recurrent or metastatic HNSCC (NCT03083873). The combination of lifileucel with pembrolizumab appears more promising based on preliminary results (ORR 44% among nine patients with HNSCC); however, this study is ongoing (NCT03645928) [127]. Another autologous TIL (TBio-4101) was previously studied with pembrolizumab in advanced solid tumors, including HNSCC with only piecemeal reporting of results; however, as of early 2025, it is no longer being developed (NCT05576077) [128,129,130]. Table 2 summarizes the ongoing studies of ACT in HPV-associated HNSCC.

## 8. Peptide–HLA-Based Immunotherapeutic Platform

A number of manufacturing, safety, and logistical considerations represent potential limitations to current ACTs. In this context, the Immuno-STAT^TM^ (Selective Targeting and Alteration of T-cells) framework is an off-the-shelf, peptide–HLA-based immunotherapeutic platform designed to activate antitumor T-cells in vivo in an APC-independent manner. These constructs consist of a covalent fusion of a MHC class I allele, target peptide epitope, co-modulator(s), and an Fc domain [131]. Specifically, CUE-101 is an Immuno-STAT^TM^ construct being actively investigated in HPV-16-positive HNSCC, consisting of an HLA complex (HLA-A*0201), an HPV-16 E7 peptide epitope, and four attenuated IL-2 molecules [132]. Interim results from an ongoing phase I study in HPV-16-positive recurrent/metastatic HNSCC demonstrated an ORR of 47% and median PFS of 5.8 months for the combination of CUE-101 and pembrolizumab among 19 evaluable PD-L1-positive patients in the first-line setting. Among 19 evaluable patients receiving CUE-101 monotherapy in the subsequent-line setting after previous first-line platinum-based chemotherapy and/or pembrolizumab, there was a single partial response (ORR 5%). CUE-101 appeared well tolerated both as a monotherapy and in combination with pembrolizumab, with 92% of AEs of grade 2 or less [133].

## 9. Conclusions and Future Directions

The underlying molecular biology, TME, and host immune response in HPV-associated HNSCC offers several potential vulnerabilities for therapeutic targeting. Checkpoint inhibitor immunotherapy, specifically PD-1 inhibition, has been standardly applied to advanced HNSCC regardless of HPV status. Yet only a subset of patients derives durable benefit, and there is strong preclinical evidence to support immune evasion and local immunosuppression in HPV-related oncogenesis. Therapeutic vaccines, TCR T-cell therapies, and peptide–HLA-based constructs represent emerging immunotherapeutic strategies in this space, nearly all of which target HPV E6 or E7 oncoproteins. Response rates to these therapies have been modest thus far, though they appear more promising when combined with checkpoint inhibitor immunotherapy. The restriction of TCR T-cell therapies and peptide–HLA-based constructs to a few HLA alleles limits the candidate population, and further understanding of resistance via genetic alterations in antigen processing is needed.

Much of our understanding of oncogenesis in HPV-associated HNSCC is derived from primary tumors in patients with localized disease. Several recent studies have identified molecular alterations characteristic of recurrent but not primary tumors, some of which overlap with HPV-negative HNSCC [134,135,136]. Nearly all the aforementioned novel immunotherapies are primarily being studied in the recurrent or metastatic setting. Whether recurrent disease relies upon the same viral oncogenic machinery and immune environment as in localized disease requires further study, which may help guide the appropriate setting in which to apply these therapies [137].

Despite our detailed understanding of the essential HPV-associated oncoproteins, no small molecule inhibitors have thus far emerged as potential therapies in the clinic. Limitations to this development have included the small molecular size of these proteins, their structural similarity to functional domains of proteins required for normal function, and the lack of enzymatic activity (other than E1) targeted by currently available antiviral agents [90,138]. Emerging strategies with early in vitro data from cervical cancer cell lines include the prevention of downstream tumor suppressor degradation via targeted disruption of viral oncoprotein binding (e.g., between E6/p53 or E7/PTPN14) and targeted HPV oncoprotein degradation via proteolysis-targeting chimera (PROTAC) technology [138,139,140,141]. Other candidate strategies with preclinical evidence includes siRNA and short hairpin RNA (shRNA) to silence E6, E7, or the LCR of the HPV genome, CRISPR/Cas9 engineering to target HPV E6 or E7, and a small molecule compound capable of inhibiting the ATPase activity of the HPV E1 helicase [142,143,144,145,146,147,148]. Despite these emerging avenues of development, further translational efforts are needed to therapeutically exploit the viral machinery of HPV-associated HNSCC, as well as other virus-driven malignancies.

## Figures and Tables

**Table 1 viruses-17-00712-t001:** Ongoing clinical trials of therapeutic vaccines in HPV-associated HNSCC.

NCT Number	Design	Population	Treatment	Vaccine Target	Primary Outcome	Status *
**DNA-based vaccines**						
NCT06016920	Phase I/II, multi-center, non-randomized, *n* = 51	Previously untreated recurrent/metastatic HPV-16-positive HNSCC with PD-L1 CPS ≥ 1	VB10.16 with pembrolizumab	HPV-16 E6/E7	Phase I: DLT, AEs Phase II: ORR, immune response	Recruiting
**RNA-based vaccines**						
NCT04534205 (AHEAD-MERIT)	Phase II/III, multi-center, randomized, *n* = 350	Previously untreated recurrent/metastatic HPV-16-positive HNSCC with PD-L1 CPS ≥ 1	BNT113 with pembrolizumab vs. pembrolizumab monotherapy	HPV-16 E6/E7	Part A: AEs Part B: OS, PFS	Recruiting
**Peptide-based vaccines**						
NCT04260126 (VERSATILE-002)	Phase II, multi-center, multi-cohort, non-randomized, *n* = 95	Recurrent/metastatic HPV-16-positive HNSCC, PD-L1 CPS ≥ 1 if checkpoint inhibitor-naïve, any PD-L1 CPS if checkpoint inhibitor-experienced	PDS0101 with pembrolizumab	HPV-16 E6/E7	ORR	Active, not recruiting
NCT06790966 (VERSATILE-003)	Phase III, multi-center, randomized, *n* = 351	Previously untreated recurrent/metastatic HPV-16-positive HNSCC with PD-L1 CPS ≥ 1	PDS0101 with pembrolizumab vs. pembrolizumab monotherapy	HPV-16 E6/E7	OS	Recruiting
NCT05232851	Phase I/II, single-center, multi-arm, non-randomized, *n* = 24	Locally advanced OPSCC with high-risk HPV-specific testing and at least one risk factor	PDS0101 alone or with pembrolizumab	HPV-16 E6/E7	Pathologic and HPV cell-free tumor DNA response	Recruiting
NCT03669718 (OpcemISA)	Phase II, multi-center, randomized, *n* = 198	Recurrent/metastatic HPV-16-positive OPSCC with PD-L1 CPS ≥ 1, anti-PD-1 therapy-naïve, first- or second-line setting	ISA101b with cemiplimab vs. cemiplimab monotherapy	HPV-16 E6/E7	ORR, AEs	Active, not recruiting
NCT04398524	Phase II, multi-center, single-arm, *n* = 65	Recurrent/metastatic HPV-16-positive OPSCC with progression on prior anti-PD-1 therapy	ISA101b with cemiplimab	HPV-16 E6/E7	ORR	Active, not recruiting
**Viral vector-based vaccines**						
NCT04180215	Phase I/II, multi-center, multi-cohort, non-randomized, *n* = 200	Recurrent/metastatic HPV-16-positive HNSCC eligible to receive pembrolizumab as standard of care	HB-201 alone or HB-201 and HB-202 combination, with standard-of-care regimen including pembrolizumab	HPV-16 E6/E7	Phase I: DLT, RP2D Phase II: ORR	Active, not recruiting
NCT05108870	Phase I/II, single-center, randomized, *n* = 98	Locally advanced HPV-16-positive OPSCC, cT3-T4 (any N), or cN1-N3	Neoadjuvant HB-201 alone vs. HB-201 and HB-202 combination, both with carboplatin and paclitaxel chemotherapy prior to definitive therapy	HPV-16 E6/E7	Phase I: DLT, AEs Phase II: Deep response rates	Recruiting
NCT04432597	Phase I/II, single-center, multi-arm, non-randomized, *n* = 70	Locally advanced or metastatic HPV-associated cancer (Phase I); stage II–III p16-postive OPSCC (phase II)	PRGN-2009 alone or with bintrafusp alfa (M7824)	HPV-16 and HPV-18 E6/E7	Phase I: RP2D, safety Phase II: Increase in CD3+ tumor-infiltrating T-cells	Active, not recruiting
NCT05996523	Phase II, single-center, single-arm, *n* = 29	Stage I-II p16-positive OPSCC planned for definitive surgery or chemoradiation	PRGN-2009 with pembrolizumab prior to definitive therapy	HPV-16 and HPV-18 E6/E7	Increase in CD3+ tumor-infiltrating T-cells	Recruiting
NCT06223568	Phase II, single-center, multi-arm, randomized, *n* = 70	Resectable stage I-II OPSCC, HPV-associated with any high-risk serotype	Neoadjuvant cisplatin and docetaxel chemotherapy alone or with PRGN-2009 prior to surgery	HPV-16 and HPV-18 E6/E7	pCR rate	Recruiting
NCT06319963	Phase I/II, multi-center, multi-cohort, non-randomized, *n* = 72	Arm A: Recurrent/metastatic HPV-associated OPSCC or cervical cancer Arm B: Untreated, locally advanced HPV-associated OPSCC (cT1-2N2-3, cT3-4N0-3) or cervical cancer (Stage IB-IVA)	Lenti-HPV-07	HPV-16 and HPV-18 E6/E7	AEs, optimal biological dose	Recruiting

* Status as of 18 April 2025. Abbreviations: AE, adverse event; DLT, dose-limiting toxicity; DNA, deoxyribonucleic acid; HNSCC, head and neck squamous cell carcinoma; HPV, human papillomavirus; OPSCC, oropharyngeal squamous cell carcinoma; ORR, objective response rate; OS, overall survival; PD-1, programmed cell death protein 1; PD-L1, programmed cell death ligand 1; pCR, pathologic complete response; PFS, progression-free survival; RNA, ribonucleic acid; RP2D, recommended phase II dose.

**Table 2 viruses-17-00712-t002:** Ongoing clinical trials of adoptive cell therapies in HPV-associated HNSCC.

NCT Number	Design	Population	Treatment	Primary Outcome	Status *
**Genetically modified TCR therapies** (all with HPV- and HLA-specific enrollment criteria)					
NCT05639972	Phase I/II, single-center, single-arm, feasibility, *n* = 15	Locoregionally advanced HPV-16-positive cancer and HLA-A*02:01 allele	E7 TCR T-cells	Feasibility of administration	Recruiting
NCT05686226	Phase II, single-center, single-arm, *n* = 20	Recurrent/metastatic HPV-16-positive cancer and HLA-A*02:01 allele	E7 TCR T-cells	ORR	Recruiting
NCT05973487	Phase I, multi-center, non-randomized, multi-cohort, *n* = 840	Unresectable/metastatic solids tumors; HPV-16-positive with HLA-A*02:01 allele in cohort C; includes additional combination cohorts with a second target antigen	E7 TCR T-cells as monotherapy and in combination with other TCRs targeting MAGE-A1, MAGE-A4, MAGE-C2, PRAME	Safety, RP2D	Recruiting
NCT02858310	Phase I/II, multi-center, single-arm, *n* = 180	Recurrent/metastatic HPV-16-positive cancers with HLA-A*02:01 allele	E7 TCR T-cells	Phase I: Safety Phase II: ORR	Recruiting
NCT05787535	Phase I, single-center, single-arm, *n* = 17	Recurrent/metastatic HPV-18-positive solid tumors and HLA-DRB1*09:01 allele	HRYZ-T101 TCR T-cells	DLT, AEs	Recruiting
NCT05952947	Phase I, multi-center, single-arm, *n* = 32	Recurrent/metastatic HPV-18-positive solid tumors and HLA-DRB1*09:01 allele	HRYZ-T101 TCR T-cells	DLT, AEs	Recruiting
**CAR-T or CAR-NK cell therapies**					
NCT06096038	Phase I/II, single-center, single-arm, *n* = 33	Recurrent/metastatic HNSCC	Autologous CAR-T against CSPG4	AEs	Recruiting
NCT04119024	Phase I, multi-center, single-arm, *n* = 18	Solid tumors expressing IL13Ralpha2	Autologous CAR-T against IL13Ralpha2	DLT, AEs	Recruiting
NCT05239143	Phase I, multi-center, multi-cohort, non-randomized, *n* = 180	Advanced/metastatic epithelial-derived cancers	Allogeneic CAR-T against MUC1-C	DLT, AEs, ORR	Active, not recruiting
NCT06682793 (DENALI-1)	Phase I/II, multi-center, single-arm, *n* = 240	Unresectable/metastatic solid tumors expressing EGFR and with loss of HLA-A*02 expression; requires germline HLA-A*02 heterozygosity	Allogeneic logic-gated Tmod^TM^ CAR-T (A2B395)	Phase I: DLT, AEs, RP2D Phase II: ORR	Not yet recruiting
NCT06383507	Phase I, single-center, single-arm, *n* = 18	Relapsed/refractory CD70 positive solid tumors	Allogeneic CAR-T against CD70	AEs	Recruiting
NCT04847466	Phase II, single-center, single-arm, *n* = 55	Recurrent/metastatic gastric or head and neck cancer with prior chemotherapy and anti-PD-1 therapy	Allogeneic CAR-NK cells against PD-L1, combined with pembrolizumab	ORR	Recruiting
**TIL therapies**					
NCT03645928 (IOV-COM-202)	Phase II, multi-center, multi-cohort, non-randomized, *n* = 245	Advanced melanoma, HNSCC, and NSCLC; recurrent/metastatic HNSCC in cohort 2A without prior checkpoint inhibitors	Autologous TIL (LN-145, lifileucel) combined with pembrolizumab	ORR, grade ≥ 3 AEs	Recruiting
NCT05902520	Phase I, single-center, multi-arm, randomized, *n* = 18	Unresectable/metastatic solid tumors	Autologous double-positive (CD39+, CD103+) CD8+ TIL (AGX148) with or without siRNA PD-1 modulation	AEs	Recruiting
NCT06236425	Phase I, single-center, multi-arm, non-randomized, *n* = 15	Recurrent/metastatic HNSCC with previous progression on pembrolizumab or pembrolizumab/platinum chemotherapy	Autologous TIL leveraging single-cell sorting of patient-specific neoepitope-reactive T-cells (TBio-4101), combined with pembrolizumab	AEs	Recruiting

* Status as of 18 April 2025. Abbreviations: AE, adverse event; CAR, chimeric antigen receptor; DLT, dose-limiting toxicity; EGFR, epidermal growth factor receptor; HLA, human leukocyte antigen; HNSCC, head and neck squamous cell carcinoma; HPV, human papillomavirus; MAGE, melanoma antigen gene; NSCLC, non-small cell lung cancer; ORR, objective response rate; PD-1, programmed cell death protein 1; PD-L1, programmed cell death ligand 1; PRAME, preferentially expressed antigen in melanoma; RP2D, recommended phase 2 dose; siRNA, small interfering ribonucleic acid; TCR, T-cell receptor.

## Data Availability

Not applicable.

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
