# Peer review of "Update: Immunotherapeutic Strategies in HPV-Associated Head and Neck Squamous Cell Carcinoma"

_viruses, 2025, doi:10.3390/v17050712_

Round 1
Reviewer 1 Report
Comments and Suggestions for Authors
This review of immunotherapeutic strategies for HPV-associated head and neck cancers is excellent. The senior author is an expert on the topic. The review is comprehensive and exceptionally well written. The tables are informative.
One tiny suggestion: In the section on TCR engineering, the E6 study from NCI is cited, but the E7 study was more promising, which is why the current trials all focus on E7. Would suggest summarizing the initial E7 study instead (or discussing both):
Nagarsheth NB, Norberg SM, Sinkoe AL, Adhikary S, Meyer TJ, Lack JB, et al. TCR-engineered T cells targeting E7 for patients with metastatic HPV-associated epithelial cancers. Nat Med. 2021;27(3):419-25.
Author Response
- Comment 1: One tiny suggestion: In the section on TCR engineering, the E6 study from NCI is cited, but the E7 study was more promising, which is why the current trials all focus on E7. Would suggest summarizing the initial E7 study instead (or discussing both): Nagarsheth NB, Norberg SM, Sinkoe AL, Adhikary S, Meyer TJ, Lack JB, et al. TCR-engineered T cells targeting E7 for patients with metastatic HPV-associated epithelial cancers. Nat Med. 2021;27(3):419-25.
- Response 1: We agree with this suggestion, and have updated the text as follows on page 10, including 2 additional references (the work by Nagarsheth et al as suggested, as well as an additional reference highlighting possible reasons for more encouraging efficacy in early studies of E7 TCR T-cells): “Potentially more promising was a phase I study of E7 TCR T-cells in a similar patient population, conducted by the same group. This product resulted in objective responses in 6 of 12 patients, including 4 of 8 with anti-PD-1 refractory disease and 2 of 4 with primary head and neck tumors [122]. The phase II component of this study is ongoing (NCT02858310). In general, most ongoing TCR studies focus on E7 TCR T-cells, which in preclinical models have demonstrated higher functional avidity and improved effector function compared to E6 TCR T-cells [123].”
Reviewer 2 Report
Comments and Suggestions for Authors
Sun and Colevas provide a review of the literature on immune therapies for HPV-associated HNSCC. They review the fundamentals of OPSCC related to HPV infection including viral pathogenesis, as well as highlighting some reported mechanisms of immune evasion in this disease. The focus is on immunotherapies for HPV-related HNSCC including anti-PD1 drugs, vaccines and adoptive T cell therapies. The authors have an excellent track record of publication in this space, the review is immaculately written, and appropriate literature are cited. This review will benefit readers of Viruses as an overview of an important field which is developing rapidly. A few minor suggestions to hopefully benefit the final draft:
- The HPV molecular biology section is extensive; these areas have been exhaustively reviewed elsewhere. The authors may consider shortening this section in favor of expanding the section regarding immune evasion/TME.
- In the section regarding checkpoint immunotherapy, there is focus on the question of whether HPV+ or HPV(-) patients are more likely to respond to anti-PD1 therapies with wording seeming to suggest a trend toward more immunogenicity of HPV+ tumors. As the authors describe, these comparisons are difficult; notably in KN-048 at 4-year follow-up there was no survival difference though there did appear to be a trend toward more benefit in HPV(-) in the forest plots. Ultimately, these may be considered different diseases and the comparison between them could be de-emphasized except to point out, as the authors do, that both can respond and in both cases there is ample room for improvement in our therapies.
- In analyzing vaccine therapies, which have to date shown limited success, the authors may consider discussing the choice of vaccine antigen (E5/6). It would be useful to cite translational studies suggesting that E2/E5 may be fruitful antigens to target (Eberhardt et al, Nature 2021).
Kudos to the authors on a well-written manuscript.
Author Response
- Comment 1: The HPV molecular biology section is extensive; these areas have been exhaustively reviewed elsewhere. The authors may consider shortening this section in favor of expanding the section regarding immune evasion/TME.
- Response 1: We appreciate the reviewer’s detailed reading of the manuscript, including this section. In response, we have consolidated the initial portion of paragraph 1 in the section “Molecular Biology of HPV-Associated Carcinogenesis” to remove the information less relevant to mechanisms of HPV-specific therapeutic targeting, as later discussed. We favor keeping the remaining text as written in order to provide the reader direct background with which to understand the later-discussed choices of existing immunotherapeutic strategies (particularly therapeutic vaccines and engineered TCRs) as well as the unmet need / potential of small molecule inhibitors we address in the “Conclusions and Future Directions” section.
- Comment 2: In the section regarding checkpoint immunotherapy, there is focus on the question of whether HPV+ or HPV(-) patients are more likely to respond to anti-PD1 therapies with wording seeming to suggest a trend toward more immunogenicity of HPV+ tumors. As the authors describe, these comparisons are difficult; notably in KN-048 at 4-year follow-up there was no survival difference though there did appear to be a trend toward more benefit in HPV(-) in the forest plots. Ultimately, these may be considered different diseases and the comparison between them could be de-emphasized except to point out, as the authors do, that both can respond and in both cases there is ample room for improvement in our therapies.
- Response 2: Thank you for this suggestion. The intent of this paragraph was to highlight that numerical trends favoring HPV-associated OPSCC have not demonstrated statistical significance relative to HPV- disease. We are glad that the reviewer shares the same general takeaway that the data are not definitive for any significant difference in anti-PD-1 response between HPV+ and HPV- OPSCC. To emphasize the reviewer’s highlighted point, we have added the following sentence on page 5 at the end of the reference paragraph: “Ultimately, PD-(L)1 inhibition has demonstrated benefit in both entities, which are biologically and molecularly distinct as previously emphasized.”
- Comment 3: In analyzing vaccine therapies, which have to date shown limited success, the authors may consider discussing the choice of vaccine antigen (E5/6). It would be useful to cite translational studies suggesting that E2/E5 may be fruitful antigens to target (Eberhardt et al, Nature 2021).
- Response 3: We agree with reviewer’s suggestion and have incorporated this point, as well as 2 additional references (from different groups) supporting a similar conclusion. The following text has been added to page 8: “Multiple independent groups have demonstrated HPV-specific T-cell responses against epitopes derived from HPV E2 and E5 proteins (in addition to E6 and E7), providing other candidate targets for therapeutic HPV vaccination [115-117].”